# Perspectives of frontline health workers on transition from development assistance for health in Ghana: A qualitative study

**Sandra Appiah-Kubi**[1], **Wenhui Mao**[2], **Augustina Koduah**[3], **Genevieve Cecilia Aryeetey**[1], **Osondu Ogbuoji**[2], **Justice Nonvignon**[1] *

**1** Department of Health Policy, Planning and Management, School of Public Health, University of Ghana, Accra, Ghana, **2** Center for Policy Impact in Global Health, Duke Global Health Institute, Duke University, Durham, NC, United States of America, **3** Department of Pharmacy Practice and Clinical Pharmacy, School of Pharmacy, University of Ghana, Accra, Ghana

* jnonvignon@ug.edu.gh

**Data Availability Statement:** This study generated qualitative data from a small sample of participants in specified institutions. Even without personal

## Abstract

Many Low-income countries depend on development assistance for health (DAH) to finance the health sector. The transition of these countries to middle-income status has led to reduction in effective aid from development partners while these countries are expected to graduate from global funding agencies such as Gavi the vaccine alliance, with implications for service delivery. The aim of this study was to explore the perspectives of frontline health workers regarding the implications of Ghana's transition to middle-income status on service delivery, the likely impact and opportunities it presents to the country. This exploratory qualitative study employed in-depth interviews to collect data from 16 health workers at three hospitals in the Greater Accra Region; one at the regional level and two at the district level. The study was conducted from December 2019 to July 2020. Data from interviews were transcribed, coded and analysed using thematic analysis in NVivo Qualitative Analysis Software version 12. The level of awareness among frontline workers about the transition and decline in DAH was generally low. Nonetheless, frontline health workers perceived that the country seems inadequately prepared for transition as donors continue to be major financiers for the sector and even for emergencies such as the current COVID-19 global pandemic. Potential challenges facilities would face due to transition may include difficulty in funding health programs, human resource challenges and delays in logistics and medicines. The implications for these will be poor health outcomes, defective monitoring and evaluation, and lapses in training programs. In addition, the perceived barriers to transition identified were poor management of resources, political interference and lack of technical expertise. While opportunities such as improvement of the health sector prioritization and efficiency, private sector involvement and autonomy could be gained. Gaps in the health intervention monitoring resulting from DAH transition could pose affect health outcomes, particularly in respect of HIV, tuberculosis and malaria. The country's preparedness to transition from DAH could be better improved with development of a clear transition plan agreed by stakeholders, including government and in-country development partners. For the health

identifiers, it is possible to expose the identity of participants based on their roles if the data are made publicly available. Besides, the IRB submission committed to not share data publicly, but that data would be held by study investigators. Data could be accessed from the following co-authors upon request: Sandra Appiah Kubi - ekuaak@gmail.com Genevieve Cecilia Aryeetey - gcaryeetey@ug.edu.gh Augustina Kodua - akoduah@ug.edu.gh Justice Nonvignon - jnonvignon@ug.edu.gh GCA, AK, JN are permanent faculty of the University and assure longer term storage and availability of data. Specific data requests will be considered, and data shared will be anonymized (without personal and institutional identifiers to protect the identity of participants, based on conditions of IRB approval. Although the authors cannot make their study's data publicly available at the time of publication, all authors commit to make the data underlying the findings described in this study fully available without restriction to those who request the data, in compliance with the PLOS Data Availability policy. For data sets involving personally identifiable information or other sensitive data, data sharing is contingent on the data being handled appropriately by the data requester and in accordance with all applicable local requirements.

**Funding:** The authors received no specific funding for this work.

**Competing interests:** The authors have declared that no competing interests exist.

sector, the eligibility for DAH transition should not simply be based on economic growth, but importantly on a country's ability to sustain ongoing and upcoming health programs.

## Introduction

Since the early 2000s, Ghana has seen an improvement in general health outcomes, partly due to global efforts such as the Millennium Development Goals and Sustainable Development Goals. However, the country continues to experience a double burden of diseases, with an increase in non-communicable diseases. For instance, in 2019, ischemic heart disease, stroke and diabetes were fourth, fifth and tenth on the top ten causes of death [1]. The ideal goal of Development Assistance for Health (DAH) is not to establish perpetual prominence or ad hoc relief to countries, but rather to support effective scale up of locally owned, operated and funded health programs incorporated within the health system of developing countries [2]. It is estimated that sub-Saharan Africa receives the largest share of DAH, with the largest share going to low-income countries [3]. In 2009, developing countries constituted 84% of the world's population and 92% of the burden of disease, but just 29% of Gross Domestic Product and 16% of health expenditure [4]. In 2016, DAH accounted for 34.6% of total health expenditure in low-income countries where significant amount of DAH is required [5].

The transition of countries from low-income to middle-income status has often come with reductions in DAH [6, 7]. Funders such as the Global Fund has considered about 11 countries excluded for subsequent HIV support based on their financial status and illness burden [8]. For many decades, development assistance for health (DAH) has been an important part of health financing in Ghana. Between 1995 and 2010, for instance, Ghana recorded about 17 new donors [9]. Since Ghana transitioned to a middle-income status in 2010, however, Gross Domestic Product (GDP) per capita has been on an upward trend and is estimated at USD 2,200 per capita in 2019 [10]. In 2010, 15 donors provided an estimated US$300 million in aid to Ghana, with Family Planning, HIV/AIDS, immunization, malaria, tuberculosis and Community Based Health Planning and Services (CHPS) programs receiving the largest share [11]. External funds from all donors comprised 20% of total health expenditure between 2012 and 2017 [12]. Donor agencies like DANIDA have ended direct budget support [13], and the country is projected to graduate from Gavi, the vaccine alliance in about 2025 [12]. A central concern during this transition is how to sustain public health interventions such as immunizations, family planning, HIV, tuberculosis and malaria programs which are heavily donor funded. Exits can also affect the health system through service delivery, medicines and technologies, human resources for health, and health financing [7]. The transition from donor aid to self-financing may influence the provision of essential public health services funded or co-funded by support. Public health achievements attained by beneficiary countries seem to be at risk unless the donor aid transition is well planned and executed. For instance, Flanagan et al. [14] show that donor transition had a negative impact on HIV service delivery in Romania and argue that countries such as Nigeria and Cambodia could face challenges sustaining HIV programs beyond transition [13]. Similarly, Wilhelm et al. [15] find that in Uganda, PEPFAR transition from direct support to health facilities discontinuing outreach, reduction in care access, and quality of HIV services.

The perspectives of country level stakeholders on DAH transition and impacts of service delivery is crucial for understanding a country responds or takes advantage of DAH transition to sustain service delivery. A previous study by Mao et al. [6] examined the perspectives of

mainly national-level policymakers, donors, civil society representatives and higher-level practitioners on DAH transition. However, the perspectives of frontline health workers on DAH transition is crucial for national level planning, since they serve as interface with clients and would better appreciate the impact of transition on service delivery. Therefore, the aim of this study was to explore the perspectives of frontline workers on Ghana's transition from DAH, the challenges and opportunities presented by the transition, as well as the perceived impact on service delivery and the perceived strategies for managing the transition.

## Methods

### Study design

This study used an exploratory qualitative design to explore the perspective of frontline health workers on DAH transition.

### Study setting

Three hospitals in the Greater Accra region of Ghana were purposively selected for the study. These hospitals (like others) receive funding from development partners through the Ministry of Health and Ghana Health Service to aid in implementing public health programs. The Greater Accra Regional Hospital located in Accra the capital city of Ghana with a bed capacity of 420, provides healthcare for an estimated population of about 4,283,322 inhabitants in the region. Tema General Hospital is the main government hospital serving the Tema metropolis, has a 294-bed capacity offering health services to an estimated population of 371,220. Ledzokuku-Krowor Municipal Assembly Hospital (LEKMA HOSPITAL) is a 100-bed capacity hospital, has a malaria research Centre and serving a population of 227,932. The Greater Accra Regional Hospital is a secondary facility serving as a referral centre for the primary facilities in the region; Tema General Hospital and LEKMA hospital inclusive. The study explored the perspectives of frontline health workers across the different facility levels.

### Sampling technique

A purposive sampling technique was used to select frontline health workers in the Greater Accra Regional Hospital, LEKMA Hospital and Tema General Hospital who have worked at the public health departments for more than two years as this may have exposed them to interactions at various meetings on financing of public health programs and activities. After identifying the initial set of participants, other health workers in the same departments were reached through snowballing. Different categories of health workers (medical doctors, nurses and disease control officers) were identified to participate in the interviews. A total of 16 frontline health workers (i.e., nine medical doctors, five nurses and two disease control officers—Table 1 provides background characteristics of study participants) were interviewed upon saturation. No identified potential participants refused to participate in the study.

### Data collection

The COVID-19 pandemic limited ability to conduct face to face interviews. Semi-structured interviews using an interview guide (S1 Text) via telephone were conducted by SAK among the 16 frontline health workers with each interview lasting between 30 to 40 minutes. The questions focused on frontline health workers' understanding of DAH for a country in transition, domestic financing for health, the potential impact of transition on service delivery and outcomes, challenges and opportunities of the transition, as well as strategies to manage donor transitions. The interviews were conducted by the same interviewer (i.e. SAK, female; MD,

**Table 1. Background characteristics of participants.**

| Participants | Sex | Age | Level of Education | Category of Staff | Facility |
|---|---|---|---|---|---|
| MD 1 | Female | 32 | Masters | Medical Doctor | TGH |
| MD 2 | Male | 34 | Masters | Medical Doctor | LEKMA |
| MD 3 | Male | 37 | Masters | Medical Doctor | GARH |
| MD 4 | Female | 29 | First Degree | Medical Doctor | GARH |
| MD 5 | Female | 34 | Masters | Medical Doctor | GARH |
| N 1 | Female | 47 | First Degree | Nurse | GARH |
| MD 6 | Female | 31 | Masters | Medical Doctor | GARH |
| MD 7 | Female | 37 | Masters | Medical Doctor | GARH |
| N 2 | Male | 35 | Diploma | Nurse | TGH |
| DCO 1 | Male | 32 | First Degree | Disease Control Officer | TGH |
| N 3 | Male | 41 | First Degree | Nurse | TGH |
| N 4 | Female | 36 | First Degree | Nurse | LEKMA |
| MD 8 | Male | 35 | Masters | Medical Doctor | TGH |
| DCO 2 | Female | 49 | First Degree | Disease Control Officer | LEKMA |
| N 5 | Female | 46 | Diploma | Nurse | LEKMA |
| MD 9 | Female | 32 | First Degree | Medical Doctor | LEKMA |

Notes: MD–Medical Doctor; N–Nurse; DCO–Disease Control Officer.

MPH student; prior to the study commencement, no relationship was established with participants) to ensure uniformity of data collection. The interviewer received training on qualitative research as part of MPH and before commencement of the interview to ensure uniformity of data collection. The interviews were audio recorded and notes taken with the permission of the participants. No repeat interviews were conducted. All interviews were conducted in English.

## Data analysis

All the audio recordings were transcribed verbatim, and transcripts were coded by SAK and AK and categorized into themes based on their similarities. Two members of the team reviewed transcripts to ensure that they were consistent. Transcripts were not returned to participants for comments. Thematic content analysis of the collected data was undertaken using NVivo version 12. Four broad themes were developed in advance in relation to the objectives of the study. The themes were knowledge on donor transition, donor transition challenges, donor transition opportunities and managing donor transition. Sub-themes were then deduced from the data to explain the four broad themes. The themes and related quotations presented rich and unique experiences of the participants. Reporting of the findings of this study follows the Consolidated criteria for reporting qualitative research (COREQ–S1 Checklist).

## Ethical considerations

The study received ethical approval (GHS-ERC 039/03/20) from the Ghana Health Service Ethics Review Committee. Written informed consent was sent to participants via Whatsapp messages, after which appointments were scheduled. At the time of interviews, each participant was asked to provide verbal consent via the telephone. The participants were informed that the interview was voluntary, and they had the option to withdraw from the interview process at any point of the study without any intimidation. Further, participants were assured of

confidentiality and privacy. In safeguarding identity and ensuring anonymity of the participants, code numbers (MD 1–9, N 1–5, DCO 1–2) were assigned instead of their names. The data collected were kept in confidentiality and used exclusively for the intended purpose. The electronic information obtained were stored in password protected files accessible to only the researcher and would be destroyed after five years.

## Results

### 1. Understanding of DAH and donor aid transition

Frontline health workers had similar views on donor aid transition. They perceived donor aid transition as a shift from development partner funding of public health programs to self-financing by the Government. They explained it as the gradual weaning of donor assistance with the aim of preparing the country to take over the funding of these programs completely.

Most of health workers described a successful transition as one where the donors adequately aid the recipient country with necessary skills and knowledge on how to take over funding the health programs on their own. Others perceived a successful transition as one where the country is able to sustain all health programs that were heavily donor funded once the development partners exited.

*Transition basically means the [country] has to find other sources of funding to replace what we will lose by the fact that we have moved from a low to a middle-income country. I would say its successful when we can fund our activities ourselves so once we can do that, I think we can say we have had a successful transition; so all these free things such as HIV drugs, vaccination if it comes to a point when they are not coming from donors or a facility can fund on its own, then we will say it is successful* (MD 8)

Many of the participants seemed unaware that there is an on-going transition from DAH in the health sector. Among the health workers who were aware of the on-going donor transition, a few of them indicted that they were aware of plans by the MOH to develop a roadmap involving how to finance the health sector without aid from donors.

*Yes, I am aware of the transition, but I think it hasn't been fully considered because I have heard about some organizations cutting down on the amount of money, they are giving out for certain programs that we have now. I know there are various plans and there have been several symposiums under the theme "Ghana beyond aid and our innovations" to the health sector. . ..* (MD 4)

### 2. Donor aid transition readiness

On Ghana's readiness towards donor aid transition, all the participants perceived that Ghana was not adequately prepared to transition from receiving DAH. In their view, most of the public health programs are largely donor funded and even those with major government funding have complementary support from donors. Thus, the possibility of successfully moving away from receiving aid may take a while to be effective. Some participants were of the view that health emergencies such as the current COVID-19 pandemic worsen the country's preparedness towards DAH transition.

*As a doctor who has been in the public sector for some time and interacting with patients who are in need of these services, health care delivery like TB, HIV, I would say that even in the*

*midst or in the presence of these donors we are still struggling to handle the programs we are talking about basically and I'm not sure if we are ready to take over on our own. . .* (MD 1)

## 3. Donor transition challenges

The study participants indicated that the country will likely face challenges both at the facility and national level because of donor transitions. The challenges identified were categorized into financial, human resource, logistics and medicines, and challenges with monitoring and evaluation. These are described below.

**Financial challenges.**   The participants opined that the transition would limit the amount of money available to support several programs that are usually donor sponsored. Already the health sector budget of the country is largely donor funded and may place a huge burden on the government to independently provide all the funds to support the health sector. Facilities may be required to finance some of the programs when donors leave, and government is unable to further provide support for these programs.

*I think the biggest challenge is mainly financial. . . Most health facilities would not have strong systems you know if these donor aid do not come with a certain financial component. So, for me I think that the biggest challenge will be the fact that all that financial burden will be on the facility to take up. I think that will be the biggest challenge.* (MD 3)

**Human resource challenges.**   Majority of the participants perceived that there will be challenges in terms of human resource allocation in the facilities as a result of donor transition. The donors provide human resources for example community-based volunteers for the implementation of public health programs. Exiting may result in a reduction in human resources available for the interventions. Some donors employ other personnel to support implementation of programs, sometimes at the community levels. These personnel are not within the government pool of workers and thus exit of donors means exit of these workers as government may not be able to employ them. Again, some health workers revealed that the transition could result in defect in training programs since some training programs are organized and funded by donor partners.

*You know whenever there is a public health program like this we don't only rely on the staff, we engage community-based volunteers and because funding is going to be minimized in my view, people needed to work will also be minimized, [implying that] the staff who are government workers, who are also very few, are going to perform the task of a larger number.* (DCO 1)

**Delays in logistics and medicines.**   Frontline workers expressed the view that DAH transition could leave gaps in the financing and procurement of medicines and other logistics required for effective service delivery, affecting sustainability.

*A lot of the logistics come in, [for] example the new combined antenatal weighing card which is something that we have a lot of donor support for. It is capital intensive to print one book so if you have a heavy antenatal clinic and the donor transitions [from financing] that particular program you realize that the burden of printing these antenatal books will shift to the hospitals and health centres, and the health directorate will not be able to support that because most of these institutions do not have a budget set aside to support such programs. When it*

*comes to immunizations and vaccinations, one challenge is that we run out even presently of weighing cards for populations sometimes for a whole three months.* (MD 3)

**Challenges with monitoring and evaluation.**   The participants revealed that donor-funded programs often come with strong monitoring and evaluation systems, cushioned by finances and dedicated personnel, to support success. Again, the donors in a bid to make their interventions successful, often provide a monitoring and evaluation team to ensure resources are managed effectively and efficiently to avoid wastages and produce good results. Exits may lead to failure in managing resources efficiently.

*Monitoring and evaluation of policies and projects [would be] a big challenge. When we had more people, we go round for supportive supervision and monitoring of health programs. Some groups [donors] were paying people to do various activities but now those people are not within the facility anymore because the groups are not supporting so those people are not getting paid, so they are not working.* (MD 9)

## 4. Opportunities from transition

Opportunities exist for the country to take advantage of donor transitions, as expressed by participants. Such opportunities include increased private sector involvement, improvement of health sector prioritization and efficiency, and autonomy.

**Increased private sector involvement.**   Most of the participants perceived that transiting from donor assistance provides an opportunity to engage the private sector in efforts towards improving delivery and financing of the health sector. The creation of public-private partnerships could improve the economy and, in the end, improve the health sector to provide good health services to the community.

*It creates the opportunity for private institutions to partner with governmental institution so this could improve privatization in the economy which in the long run makes the economy of Ghana much better, so it gives that opportunity for private institution to partner with government to improve service delivery.* (MD 3)

**Improved health sector prioritization.**   Transition from external aid in the health sector could be seen as an avenue for improving prioritization in the health sector in that the sector will be forced to step up and manage health programs on their own. The transition provides an opportunity for the sector to offer innovative and efficient measures for the running of health programs. It provides an avenue for increasing domestic revenue to support the health system.

*The opportunity will be for the country to explore the avenue for resource generation. And we can tell of our story. I mean the bit is proving to the world that you can do it so that you become like a learning hub. Transition also helps countries determine what their priorities are and ensure that we take up the programs and pursue the agendas that will enable us to achieve the right outcomes in terms of healthcare for a healthy nation* (MD 7)

**Autonomy.**   All health workers perceived the donor transition as creating a sense of autonomy in the health sector, thus being independent in managing of health programs and taking appropriate decisions with the country's interest in mind and not what the development partners envisaged. Donor transition offers an opportunity for the health system to build capacity, thus putting their knowledge and skills into practice.

*We are forced to learn how to depend on ourselves at least if we are able to adequately do that then we will end up with a system that supports itself and its self-maintained so that in the future with just the government support the TB program will be going on.* (MD 6)

## 5. Managing donor transitions

The study participants proposed measures to ensure a successful transition. Some of the participants were of the view that in the case of health, eligibility for DAH transition should not simply be based on economic growth, but importantly on a country's ability to sustain ongoing and upcoming health programs.

*There is no point having done the good work a donor comes to a country, you have done the good work to help a country but they are still not able to take over then there is no point because when you leave all your good work will definitely be run down so all your coverage that you have succeeded in, all your measures that you had; because donors will not leave completely they will always come back to measure outcomes, so they will always know that this thing has failed after they left.* (MD 1)

**Engagement of stakeholders in transition process.**    All participants revealed monitoring of the transition process should include the involvement of stakeholders from the national level to the periphery. These stakeholders must include representatives from each health system building block as well as level, including themselves, since they directly provide services to the community. Some health workers revealed that the clients receiving the service should also be included in the transition process and also the private sector must be involved in the decision making.

*The monitoring must involve stakeholders; all the various people in the health system building blocks. You can pick people involved in service delivery, those who represent health care workers,. . . health informatics,. . . . leadership or health care management at various levels; all those involved in lab work and pharmaceuticals all those like those in every building block. The ministry should include those of us at the facility level. They make decisions for us and expect us to implement them when they haven't involved us in the planning. They should also include the clients. Seek the challenges and problems and how they can help them* (MD 6)

**Transition plan with performance indicators.**    Participants further indicated that there should be a plan to roll out the transition. Donors through, technical assistance, should help the country develop a transition sustainability plan; performing a situational analysis to assess the key pillars and the health system building blocks. The plan should include the time the planning process should commence and the duration of the transition period. Some of the workers were of the view that planning process should commence immediately we start receiving aid from the donors. The transition process should be monitored using performance indicators. This assessment will provide knowledge on effective management of the transition process. Many participants also revealed that there should be an update of transition proceedings to workers at the facility level to create awareness of donor transitions and also, should be gradually rolled out; the process should be a systematic and stepwise approach.

*I think they should monitor the objectives of the programs and compare the results every quarter or twice a year to compare the percentage achievement of the key indicators of the programs and then decide whether support is being withdrawn from a program. you have to*

*monitor the objectives of the program whether they are still been achieved; to what extent are they being achieve in comparison to when you had the support and you can tell whether you are doing well under transition or not.* (MD 6)

**Revenue generation.**   While some frontline health workers explained that their hospitals could raise additional revenue to aid in the funding of public health programs, others did not foresee the possibility of generating revenue from the HIV, TB and child welfare clinics especially when the public knows these services are supposed to be rendered free of charge.

*We could start at looking at our IGF [internally generated funds]; what we could do with it so if we have a lab that is just looking at TB maybe we could say that let's expand it and look at other laboratory things and generate some funding for it so that the returns from these ones will now go into the vulnerable diseases. When you talk about generating revenue at the ART clinic, I think those who can afford to pay for drugs should be allowed to pay, . . . and those we can't afford we give it to them for free. At least this will bring in a little money to support the clinic.* (MD 7)

The participants revealed good leadership and governance as a strategy to be employed at the national level to aid in transitioning successfully from development assistance for health.

The country will require good, committed leaders who will prioritize the health sector and pull resources to make it a robust system.

*It all boils down to leadership and people who have a vision and also the interest. It's all about priorities. Unfortunately, our health care delivery is divided into our interest so a leader interested in TB will set an agenda to make sure TB is controlled but a leader interested in say maternal health channels energy to that. It's about time we start thinking in an integrated approach because a pregnant woman can get tb so you shouldn't look at the health delivery as vertical pillars but how are they being intertwined and how is one response going to make sure that every bit and pieces is brought to bear, and it's addressed for the holistic health of the people in the district or the facility.* (MD 7)

## Discussion

The transition from DAH is inevitable as countries develop and the question of how donor-funded programs are financed after transition becomes necessary. Essential public health interventions such as immunization, family planning, HIV/AIDS, Malaria and Tuberculosis have been financed largely by development partners. In most of the health facilities involved in this study, donors finance a considerable amount of their public health programs and some of these programs have seen a decline in funding from the development partners.

The study found that there is little awareness among participants of the country's transition from DAH. Participants indicated that the country does not seem adequately prepared to self-finance its health programs. It was found that financial and technical resources are provided by development partners for some programs and there are concerns about leaving a huge gap in the financing of public health programs leading to inadequate funding available to run these programs. Further, monitoring of health programs is done effectively in collaboration with development partners. Individuals such as data managers, surveillance managers are employed specifically for monitoring of the health intervention. With the exiting of these donors, the government may have to take up this responsibility; failure to do so results in gaps in monitoring health interventions. Participants of the study acknowledge that the country is well

equipped to generate additional resources to finance the health sector however the allocation of these resources is a huge challenge. Less privileged facilities receive less funding as compared to the privileged facilities. Additionally, there is untapped potential in the private sector, which can help sustain public health and these opportunities are missed as there is a lack of private sector engagement. The transition from external aid gives the health sector the opportunity to involve the private sector in supporting the health sector. The current analysis indicates that the country needs to leverage the full potential of the private sector.

As noted by Mao et al., transition from donor aid brings about some sense of independence and autonomy as also identified in the study [6]. The country is able to make its own decisions pertaining to what it deems appropriate for its population. It is an opportunity to aim at being financially independent, thus exploring avenues of generating additional resources to run the health sector. However, unlike the findings of Mao et al., this study observed that several participants seemed unaware of the fact that a transition in the health sector was in progress [6]. This phenomenon could be attributed to, among other things, the fact that decisions concerning health programs are usually made by the national level policymakers (who constituted their study population) with little involvement of frontline health workers who are responsible for the delivery of service at the community level; a major concern when it comes to policy implementation affecting health service delivery to the population. For a successful transition, the people involved in the process should be duly involved. Further, most participants at the facility level had little knowledge about past graduation from donor aid which is indicative that these workers who are the frontline implementors of policies are not involved in the decision making hence are capable of torpedoing transition efforts.

Furthermore, DAH transition could negatively impact procurement and supply chain systems due to transition, as donor funded health programs have market knowledge and skill in procuring medicines and logistics so to avoid shortages, which skills could elude countries during transition. For example, Gavi has access to preferential prices when purchasing vaccines for immunization programs. Once they exit the country may not have access to these privileged prices leading to shortages in vaccines for immunization [16].

Again, weak human resource planning and development practices pose risks to transition if not addressed adequately. Of particular concern is the impact of transition on human resource for health. Countries often have challenges with planning for adequate number and distribution of health professionals. The loss of external support results in gaps in staffing and technical capacity, weakening the health workforce and reducing the quality of health services [17]. The study results show that development partners temporarily employ human resources for some public health interventions. This saves the government from having to employ more workers onto its payroll. There is the possibility of weakening the human resource when the donors exit if the transition is not adequately addressed. In addition, politics plays a role in the generation and mobilization of domestic resources; deciding which resources to be allocated to the health sector and which regions receives the most resources. Countries graduating from Gavi are expected to be in a position to sustain financing of health, provided there is adequate political commitment and technical capacity to plan and manage implementation of their immunization programs to ensure favourable conditions for financial self-sufficiency in vaccines and immunization services [18].

The study further found that participants recommend that as part of managing donor transitions, pre-transition assessment of readiness is crucial, taking into consideration an eligibility criterion; monitoring of transition process thus having a transition plan, involving stakeholders in the transition process and measuring output with key performance indicators. This finding is consistent with the views of national level stakeholders as indicated by Mao et al. [6]. Resch et al. indicated that managing successful transitions requires a pre-assessment of

readiness to transition, an agreed transition plan between donors and receiving countries, a framework to monitor the transition process and mechanism to ensure accountability [2].

Participants further lamented that policies are drafted at the national level for implementation at the facility level without the involvement of the actors at the facility level. Decision makers at the national level and the management of the health facilities such as the hospital directors, heads of public health unit, health workers and the general public through the civil society organizations should be included in the transition dialogue and consequently policies drafted and implemented be made to cover them. Participants indicated that laid-down plans detailing transition timelines and implications should be provided to ensure better coordination among the country and the development partners. Again, participants indicated that the transition process should be monitored using objectives of health interventions as key performance indicators to ensure smooth taking over by the government. Program specific indicators should be used to guide the process of sustainability ensuring no abrupt discontinuation. The planning process should be initiated immediately the development partners commence funding health programs, giving prior notice of when they plan to exit with clear milestones and duration should be more than two years, ideally five years. Lack of planning will lead to unsuccessful handing over of health projects resulting in poor health outcomes.

As part of measures to mitigate the challenges associated with donor transitions at the facility level, participants highlighted the efficient use of internally generated funds to support the ongoing public health programs. A genuine concern of how the population will suddenly be asked to pay a token for services such as immunizations, HIV treatments and tuberculosis treatment which were initially given for free was raised. This will have to be voluntary and from persons who can genuinely afford. The population, however, has to be educated and sensitized on the current transition and the need to support the government transition successfully from external aid.

At the national level, polices in ensuring successful transition from external aid should aim at increasing fiscal space and allocating more funds to the health sector, rather than reallocating funds from existing projects to donor-backed programs. Some portion of additional GDP should be allocated to health programs that were previously funded by donor aid. There is also the need to learn from other countries how resources are generated, an example being the sin tax. Donors and countries need to create an enabling environment for increasing reliance on domestic financing for health programs to anticipate efficiency gains. Participants indicated that development partners need to develop polices on successful transition and put in measures to implement these policies. In 2015, Gavi put in measures to ensure a smooth transition from external to domestic funding. The measures include assessment of country readiness; the design and funding of special grants to countries to prepare the way and ease transitions; the use of flexible rules on the number of years that individual countries have to become financially self-sufficient; and securing affordable prices for vaccines for countries for several years after the transition is complete [19].

Large negative impacts could evolve from poorly executed health transitions affecting many lives. Current analysis shows that the lives of people living with HIV/AIDS could be interrupted as there will be shortages in antiretroviral drugs leading to increased morbidity and mortality. Again, interruption in vaccines delivery could cause the resurgence of vaccine preventable diseases that have been controlled almost to the level of elimination. The inability to fund the ongoing intervention leaves it hanging, leading to poor health outcomes. Participants were concerned that the general public will not have access to high quality public health services if the partners reduce their funding leading to increased financial hardship. Most public health training programs are organized and funded by the donor agencies; equipping health workers with the requisite knowledge and skills needed for the implementation of the health

intervention. There will be a deficiency when it comes to training aimed at equipping health workers to manage public health related diseases.

As a response to reduce the effects to DAH transition, participants revealed there should be strong private-public partnership; industries and companies can be empowered to acquire medicines, diagnostics and vaccines needed in the operation of the on-going public health programs thus preventing abrupt stops and breakages in the coverage. Participants also indicated that strong political will and leadership is needed for smooth transitions. The study reveals that the government needs to prioritize the use of resources in the health sector. In order to prepare for transition, there is also the need to build capacity in bridging the financial gap created by the development partners; sufficient planning, advocacy and learn from other countries that have transitioned from external aid. Government ownership of health programs is also needed to transition smoothly.

## Strengths and limitations

The study findings notwithstanding, a key limitation of the study is the exclusion of potential respondents such as supply chain managers and finance officers, who play important roles in externally funded programs and whose perspectives would have been useful for the study. However, these categories of staff were unavailable for interviewing due to their hectic schedules as a result of the COVID-19 response. Again, the restrictions posed by the COVID-19 pandemic led to interviews being conducted via telephones. The lack of in-person interactions meant that the interviewer could not observe gestures and reactions from participants, which form an important part of any qualitative research. Finally, there is a limit to generalizability of the findings from this study to other settings, for example outside Ghana. Again, since all participants are based urban parts of the Greater Accra Region of Ghana, there is a limit to generalizability of findings to frontline health workers in rural areas.

## Conclusion and policy implications

This study explored the perspectives of frontline health workers on DAH transition in Ghana. The study found a low level of awareness of DAH transition among frontline workers. Participants perceived that the country seems inadequately prepared for transition as donors continue to be major financiers for the sector and even for emergencies such as the current COVID-19 global pandemic. Potential challenges to service delivery due to DAH transition include difficulty in funding health programs, human resource challenges and delays in logistics and medicines. The implications for these could be poor health outcomes, defective monitoring and evaluation, and lapses in training programs. In addition, the perceived barriers to transition identified were poor management of resources, political interference and lack of technical expertise.

These challenges notwithstanding, countries could leverage opportunities arising from DAH, including health sector prioritization and efficiency, private sector involvement and autonomy.

There are also gaps in the health intervention monitoring and the country seems inadequately prepared to transition. It is recommended that the country works towards developing a transition plan with the involvement of stakeholders including policymakers, practitioners, civil society, and development partners to provide a framework guiding the transition process and measuring output with key performance indicators in a bid to ensure a successful transition. Future research could explore the perspectives of beneficiaries (service clients, particularly of externally funded programs) on the impact of DAH on equity and quality of service delivery.

## Supporting information

**S1 Checklist. This supplementary information provides details of how the COREQ Checklist was used in reporting qualitative findings.**
(PDF)

**S1 Text. This supplementary information provides the guide used in conducting study interviews.**
(DOCX)

## Author Contributions

**Conceptualization:** Sandra Appiah-Kubi, Justice Nonvignon.

**Data curation:** Sandra Appiah-Kubi.

**Formal analysis:** Sandra Appiah-Kubi.

**Investigation:** Sandra Appiah-Kubi.

**Methodology:** Sandra Appiah-Kubi, Justice Nonvignon.

**Resources:** Sandra Appiah-Kubi.

**Software:** Sandra Appiah-Kubi.

**Supervision:** Justice Nonvignon.

**Writing – original draft:** Sandra Appiah-Kubi, Justice Nonvignon.

**Writing – review & editing:** Sandra Appiah-Kubi, Wenhui Mao, Augustina Koduah, Genevieve Cecilia Aryeetey, Osondu Ogbuoji, Justice Nonvignon.

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
