## [Decision Letter · Decision Letter 0]

3 Nov 2021

PGPH-D-21-00176

Perspectives of frontline health workers on transition from development assistance for health in Ghana: a qualitative study

Dear Dr. Nonvignon,

Thank you for submitting your manuscript to PLOS Global Public Health. After careful consideration, we feel that it has merit but does not fully meet PLOS Global Public Health’s publication criteria as it currently stands. Therefore, we invite you to submit a revised version of the manuscript that addresses the points raised during the review process.

We look forward to receiving your revised manuscript.

Kind regards,

Tara Carney, Ph.D.

Academic Editor

Journal Requirements:

1. Please provide additional details regarding participant consent. In the ethics statement in the Methods and online submission information, please ensure that you have specified (1) whether consent was informed and (2) what type you obtained (for instance, written or verbal, and if verbal, how it was documented and witnessed).

2. Please update the completed 'Competing Interests' statement, including any COIs declared by your co-authors. If you have no competing interests to declare, please state "The authors have declared that no competing interests exist". Otherwise please declare all competing interests beginning with the statement "I have read the journal's policy and the authors of this manuscript have the following competing interests:"

3. In the online submission form, you indicated that "The data collected was kept in confidentiality and used exclusively for the intended purpose. The electronic information obtained is stored in password protected files accessible to only the researcher and will be made available upon request"

Additional Editor Comments (if provided):

This is an interesting article that will add to the literature on health services in Ghana, especially in terms of exploring perspectives of health workers themselves. There are however, a number of issues that the authors will need to address before consideration for publication. These include the following:

-The introduction seems solely based on the availability of donor assistance for healthcare, but it would strengthen the argument if additional literature was included on what are the needs in terms of key healthcare issues in Ghana, and how DAH and donor assistance could help here. There are also very limited references.

An inclusion of a table on the demographics on the included participants would be valuable.

In terms of methods, it is unclear why this study is seen as a cross-sectional study when it is qualitative research. Including the key informant interview guide in an appendix would be valuable.

The authors are also not clear about if they used any kind of software for data analysis-please confirm.

Have the COREQ guidelines been taken into account for reporting qualitative findings?

The discussion section could be more structured, and it would also be valuable to think critically about the limitations of this study.

Please consider the comments below from the reviewer:

Reviewer 1:

Thank you for the opportunity to review this manuscript (PGPH-D-21-00176)! TITLE: • The title has grammatical errors. ABSTRACT: • The authors should remove “(lower)” from the second sentence in the background, as it is confusing. • The authors only mention Gavi when speaking about global funding agencies. Is there a particular reason for this? • The authors should rephrase the aim to clarify that they are wanting to explore. Is it frontline health workers’ perspectives of the implications of Ghana’s transition to a middle-income country on service delivery? • Are the results summarised from the perspective of frontline health workers? In places, it is not clear what are the perspective of participants and what are perspectives of the authors. INTRODUCTION: • The authors should reference these sentences: “The shift of a country from low-income to a higher income bracket (such as middle-income) give development partners the indication that a country is in a better position to finance its activities, hence, a reduction in DAH. Thus, countries seeing significant economic growth are no longer eligible to receive assistance, being perceived as qualified to self-finance their health programs.” • Last paragraph: The authors should provide a stronger rationale for why the perspectives of frontline health workers are important and who can benefit from this study. It is also unclear what information on donor aid transition is inadequate, compared to what, and how it can help. METHODS: • Study design: The authors should phrase the study design as an exploratory (or descriptive) qualitative study, rather than using the term “cross-sectional”. It is also advisable for the authors to be consistent In describing the aim in this study in abstract and main text, and therefore use one term throughout, either “perspectives” of “perceptions”. • Study setting: How were the three hospitals selected and why? • Sampling technique: Can the authors clarify whether they considered which departments health workers worked in and how this could have influenced their perspectives? • Data analysis: Were the interview recording transcribed verbatim? Were any principles of scientific rigour applied, e.g. peer debriefing, audit trails etc.? RESULTS: • The results are well-described, with supporting quotes. CONCLUSION: • Do the authors have any recommendations for future research?

Reviewers' comments:

Reviewer's Responses to Questions

**Comments to the Author**

1. Does this manuscript meet PLOS Global Public Health’s publication criteria? Is the manuscript technically sound, and do the data support the conclusions? The manuscript must describe methodologically and ethically rigorous research with conclusions that are appropriately drawn based on the data presented.

Reviewer #1: Yes

2. Has the statistical analysis been performed appropriately and rigorously?

Reviewer #1: N/A

3. Have the authors made all data underlying the findings in their manuscript fully available (please refer to the Data Availability Statement at the start of the manuscript PDF file)?

Reviewer #1: No

4. Is the manuscript presented in an intelligible fashion and written in standard English?

Reviewer #1: Yes

5. Review Comments to the Author

Reviewer #1: Thank you for the opportunity to review this manuscript (PGPH-D-21-00176)!

TITLE:

•The title has grammatical errors.

ABSTRACT:

•The authors should remove “(lower)” from the second sentence in the background, as it is confusing.

•The authors only mention Gavi when speaking about global funding agencies. Is there a particular reason for this?

•The authors should rephrase the aim to clarify that they are wanting to explore. Is it frontline health workers’ perspectives of the implications of Ghana’s transition to a middle-income country on service delivery?

•Are the results summarised from the perspective of frontline health workers? In places, it is not clear what are the perspective of participants and what are perspectives of the authors.

INTRODUCTION:

•The authors should reference these sentences:

“The shift of a country from low-income to a higher income bracket (such as middle-income) give development partners the indication that a country is in a better position to finance its activities, hence, a reduction in DAH. Thus, countries seeing significant economic growth are no longer eligible to receive assistance, being perceived as qualified to self-finance their health programs.”

•Last paragraph: The authors should provide a stronger rationale for why the perspectives of frontline health workers are important and who can benefit from this study. It is also unclear what information on donor aid transition is inadequate, compared to what, and how it can help.

METHODS:

•Study design: The authors should phrase the study design as an exploratory (or descriptive) qualitative study, rather than using the term “cross-sectional”. It is also advisable for the authors to be consistent In describing the aim in this study in abstract and main text, and therefore use one term throughout, either “perspectives” of “perceptions”.

•Study setting: How were the three hospitals selected and why?

•Sampling technique: Can the authors clarify whether they considered which departments health workers worked in and how this could have influenced their perspectives?

•Data analysis: Were the interview recording transcribed verbatim? Were any principles of scientific rigour applied, e.g. peer debriefing, audit trails etc.?

RESULTS:

•The results are well-described, with supporting quotes.

CONCLUSION:

•Do the authors have any recommendations for future research?

6. PLOS authors have the option to publish the peer review history of their article (what does this mean?). If published, this will include your full peer review and any attached files.

**Do you want your identity to be public for this peer review?** For information about this choice, including consent withdrawal, please see our Privacy Policy.

Reviewer #1: No

---

## [Decision Letter · Decision Letter 1]

18 Jan 2022

PGPH-D-21-00176R1

Perspectives of frontline health workers on transition from development assistance for health in Ghana: a qualitative study

Dear Justice Nonvignon

Thank you for submitting your manuscript to PLOS Global Public Health. After careful consideration, we feel that it has merit but does not fully meet PLOS Global Public Health’s publication criteria as it currently stands. Therefore, we invite you to submit a revised version of the manuscript that addresses the points raised during the review process.

We would like to thank the authors for their revised submission, and modifications made to the article. Please note the following:

Introduction:

-the changes recommended by the editor have not been made-please see original comments. The literature is very much still based on the DAH only, and no real description of heath issues faced in the country are evident.

Method

-COREQ guidelines-the authors say that these have been included. Can more specific information be provided in the responses to confirm this, an if helpful the guidelines checklist can be included as an appendix.

-Please can you add more detail on how telephonic consent was obtained.

Discussion

-add the year published to Mao et al

-Limitations: thank you for adding some information here, but this is very brief. Think about the effect on qualitative work that telephonic interviews may have had, and the fact that the data is not generalisable.

We look forward to receiving your revised manuscript.

Kind regards,

Tara Carney, Ph.D.

Academic Editor

Journal Requirements:

1. Please ensure that you refer to Table 1 in your text as, if accepted, production will need this reference to link the reader to the table.

2. Please amend your Financial Disclosure statement. If you did not receive any funding for this study, please simply state: “The authors received no specific funding for this work.”

3. Please update your Competing Interests statement. If you have no competing interests to declare, please state: “The authors have declared that no competing interests exist.”

Additional Editor Comments (if provided):

Reviewers' comments:

Reviewer's Responses to Questions

**Comments to the Author**

1. If the authors have adequately addressed your comments raised in a previous round of review and you feel that this manuscript is now acceptable for publication, you may indicate that here to bypass the “Comments to the Author” section, enter your conflict of interest statement in the “Confidential to Editor” section, and submit your "Accept" recommendation.

Reviewer #1: All comments have been addressed

2. Does this manuscript meet PLOS Global Public Health’s publication criteria? Is the manuscript technically sound, and do the data support the conclusions? The manuscript must describe methodologically and ethically rigorous research with conclusions that are appropriately drawn based on the data presented.

Reviewer #1: Yes

3. Has the statistical analysis been performed appropriately and rigorously?

Reviewer #1: N/A

4. Have the authors made all data underlying the findings in their manuscript fully available (please refer to the Data Availability Statement at the start of the manuscript PDF file)?

Reviewer #1: Yes

5. Is the manuscript presented in an intelligible fashion and written in standard English?

Reviewer #1: Yes

6. Review Comments to the Author

Reviewer #1: The authors have done well in addressing the comments.

7. PLOS authors have the option to publish the peer review history of their article (what does this mean?). If published, this will include your full peer review and any attached files.

**Do you want your identity to be public for this peer review?** For information about this choice, including consent withdrawal, please see our Privacy Policy.

Reviewer #1: No

---

## [Editor Report · Decision Letter 2]

12 Feb 2022

PGPH-D-21-00176R2

Perspectives of frontline health workers on transition from development assistance for health in Ghana: a qualitative study

Dear Dr. Nonvignon 

Thank you for submitting your manuscript to PLOS Global Public Health. After careful consideration, we feel that it has merit but does not fully meet PLOS Global Public Health’s publication criteria as it currently stands.

Therefore, we invite you to submit a revised version of the manuscript that addresses the points raised during the review process.

I would like to thank the authors for the revisions that they have made to this manuscript, which have improved the article. However, there are still three minor areas that I would like to see some changes on, as listed below: 1) COREQ guidelines: please ensure that these match up to what is in the text of your article. There are items under relationship with participants that are marked as n/a in the guidelines, but then are explained in the text. Please rectify this. 2) informed consent: thank you for revising this. However, it is still not completely clear what happened-were participants sent a whatsapp just to read, or was there a way that they could respond in any way. if verbal consent was obtained on the phone, how was this done-did the researcher still sign a hard copy of the document? Some more detail is needed here. 3)limitations: this section is more explanatory now, but I would just like to see an extra sentence on the generalisability of the study findings-to what? other frontline health workers? in Ghana or other countries? in rural vs more urban settings? this really just requires a bit more explanation.

We look forward to receiving your revised manuscript.

Kind regards,

Tara Carney, Ph.D.

Academic Editor

Journal Requirements:

1. Please amend your Financial Disclosure statement. If you did not receive any funding for this study, please simply state: “The authors received no specific funding for this work.”

2. Please update your Competing Interests statement. If you have no competing interests to declare, please state: “The authors have declared that no competing interests exist.”

3. We have noticed that you have uploaded supporting information but you have not included a list of legends.  Please add a full list of legends for all supporting information files (including figures, table and data files) after the references list.

Additional Editor Comments (if provided):

I would like to thank the authors for the revisions that they have made to this manuscript, which have improved the article. However, there are still three minor areas that I would like to see some changes on, as listed below:

1) COREQ guidelines: please ensure that these match up to what is in the text of your article. There are items under relationship with participants that are marked as n/a in the guidelines, but then are explained in the text. Please rectify this.

2) informed consent: thank you for revising this. However, it is still not completely clear what happened-were participants sent a whatsapp just to read, or was there a way that they could respond in any way. if verbal consent was obtained on the phone, how was this done-did the researcher still sign a hard copy of the document? Some more detail is needed here.

3)limitations: this section is more explanatory now, but I would just like to see an extra sentence on the generalisability of the study findings-to what? other frontline health workers? in Ghana or other countries? in rural vs more urban settings? this really just requires a bit more explanation.
---

## [Editor Report · Decision Letter 3]

9 Mar 2022

Perspectives of frontline health workers on transition from development assistance for health in Ghana: a qualitative study

PGPH-D-21-00176R3

Dear Dr Nonvignon 

We are pleased to inform you that your manuscript 'Perspectives of frontline health workers on transition from development assistance for health in Ghana: a qualitative study' has been provisionally accepted for publication in PLOS Global Public Health.

Best regards,

Tara Carney, Ph.D.

Academic Editor

Thank you for addressing the remaining comments that the editor had.

This manuscript is now ready for publication-however, I would recommend a final read through to address minor spelling and grammatical errors.